# Physiotherapy for Cervical Dystonia: A Systematic Review of Randomised Controlled Trials

**DOI:** 10.3390/toxins14110784

**Published:** 2022-11-11

**Authors:** Dana Loudovici-Krug, Steffen Derlien, Norman Best, Albrecht Günther

**Affiliations:** 1Institute of Physiotherapy, Jena University Hospital, 07740 Jena, Germany; 2Hans-Berger-Department of Neurology, Jena University Hospital, 07740 Jena, Germany

**Keywords:** physical therapy, rehabilitation, spasmodic torticollis, add-on treatment

## Abstract

Physiotherapy is mentioned as an adjunctive treatment to improve the symptoms of cervical dystonia in terms of pain, function and quality of life. However, botulinum neurotoxin injection remains the treatment of choice. This systematic review emphasizes physical therapy and evaluates it by including six studies. The methodology is based on a previous systematic review on this topic to provide better comparability and actuality. For this purpose, two databases were searched using the previously published keywords. This time, only randomised controlled trials were evaluated to increase the power. In conclusion, additional physical therapy and active home exercise programs appear to be useful. Further research should focus on the dose–response principle to emphasize physical therapy treatment modalities.

## 1. Introduction

Dystonia is characterised by involuntary contractions that lead to abnormal postures or movements [1]. In particular, cervical and neck muscles are affected in cervical dystonia (CD) [2].

Selective peripheral denervation of the affected muscle groups by local injection of botulinum neurotoxin (BoNT), usually, is the therapy of first choice in CD [3]. Nevertheless, additional physiotherapy or physical therapy, in addition to regular BoNT therapy, is highlighted in the German S1-guideline [4].

In 2014, De Pauw et al. published a systematic review on the topic of physiotherapy (PT) for CD with first information on PT as an adjuvant therapy option, next to treatment with BoNT. However, due to the lack of high quality studies, no recommendation could be drawn at this time, although there are beneficial effects on pain and disability or quality of life [5]. For this reason, a further updated systematic review, only on randomised controlled trial, fills the gap.

Hence, the full review question aimed at the effects of additional physiotherapy for patients suffering from CD and treated with BoNT, compared to medical therapy alone, sham therapy or primary passive therapies.

## 2. Methods

This systematic review was registered at the PROSPERO register with the number CRD42022333352 [6]. It is reported using the PRISMA guidelines [7].

Eligibility criteria—The search of relevant literature followed the PICO Framework:

(P) Patients with idiopathic CD;

(I) Physiotherapy treatment;

(C) Compared to control or alternative treatment;

(O) Pain, disability, function or quality of life.

The study design of a randomised controlled trial (RCT) was determined. The prerequisite was always the basic treatment by means of BoNT therapy.

Information source—The search strategy for the previous literature was based on an article by De Pauw et al. [5]. This was the logical consequence, as the content revolved around the same subject matter, and it also allowed for better comparability between the two literature searches.

In contrast to the previous systematic review, only RCTs were included in the present study to enable stronger recommendations from the results.

PubMed and Web of Science were searched through March 2022 to identify relevant articles. The following key words were used for both databases:Cervical dystonia OR spasmodic torticollis

AND

Physiotherapy OR rehabilitation OR physical therapy OR stretching OR relaxation OR paramedical treatment OR manipulation OR relaxation therapy OR neuromotor rehabilitation OR training OR exercise therapy.

The found hits were selected if it was a physiotherapeutic intervention study without further pharmacological context in patients with CD in a controlled setting. The language was limited to English and German.

Data collection—Data extraction was conducted by two people. This included the number of patients, their CD duration and severity, and the medication. Furthermore, the intervention and control were stated, with the belonging measuring time points and at least the results of the study including comparative values are presented. Uncertainties were discussed among the researcher team in order to achieve a consensus.

Study quality and risk of bias—The PEDro scale was used to illustrate the methodological quality of the discussed RCTs. It is now increasingly being used to rate clinical trials included in systematic reviews [8]. The improved Risk-of-bias-Tool2 (RoB2) supported the assessment of the bias through the studies concerning the Toronto Western Spasmodic Torticollis Rating Scale (TWSTRS) [9]. The advantage of the TWSTRS is the combination of functional CD features concerning severity as well as patient perceived matters of importance such as disability and pain [10]. Disagreements regarding the evaluation scales were resolved by a third opinion.

## 3. Results

Study selection—The search via PubMed revealed 29 hits referring to RCT, the search via Web of Science 132 more. However, 22 resp. 125 other articles were excluded by title or abstract for reasons like pharmacological context or missing physiotherapy or even a control group. Four duplicates were deleted. Finally, four more articles were excluded because of missing concomitant BoNT-Therapy. Therefore, six studies could be included in this review (Figure 1).

Study characteristics—The characteristics and main outcome of the included six RCTs are summarised in Table 1. The color is for better recognition of the column heading.

Even due to the small number of RCTs, there were deviating therapy models used and investigated:

BoNT alone compared to BoNT plus PT [11,12,13];

BoNT plus relaxation compared to BoNT plus PT [14];

BoNT plus regular PT compared to BoNT plus specialised PT [15];

BoNT plus PT compared to PT alone [16].

As an aside, this heterogeneity is the reason why it is impossible to conduct a meta-analysis at this time.

Although the BoNT therapy was individually adapted, the principal application was similar. However, the PT of the included trials differed quite more. It consisted of massage, stretching, postural re-education, muscle strengthening and biofeedback as well as isometric and active exercises or even taping. These therapeutic applications were conducted in a regular or especially emphasised manner. Moreover, it can be recommended as a home program or supervised by a physiotherapist. Even the length differed from 30 up to 90 min for each therapy session. At least the number of PT session varied from 8 to more than 40.

In fact, the research results presented here state that there were significant improvements for BoNT plus PT for patients with CD, but there seems to be no clear difference between several physical interventions.

Study quality—The methodological quality was good for all of the included studies, because by means of the PEDro evaluation six up to eight points were reached each (Table 2). Mainly, the missing points were with regard to the blinding of the patients and therapists and the lack of the concealed allocation of the randomisation. The risk of the bias tool allowed for the assessment of one study as being of low risk [15]. The other five RCTs were with some concerns towards the risk of bias, which means the randomisation process, the selection of the reported results or deviations from the intended intervention (Table 3). Here, also, the concealed allocation and missing blinding reduced the evaluable quality [11,12,13,16]. The concerns regarding the selection of the reported results were due to the lack of publication of the study protocol [11,13,16]. One patient dropped out for missing the interventional benefit; however, the analysis was due to intervention to treat [14].

## 4. Discussion

It is difficult to make a clear PT recommendation based on the individual six RCTs. The study by Stankovic et al. at least confirms the importance of BoNT therapy as an integral part of the therapy plan in CD [16]. It is the only study that tests PT plus BoNT against PT alone. The therapy’s success occurs mainly for patients on the combination therapy. In the other studies, significant improvements to the baseline were measured in most cases but not always between the intervention and control groups [15]. However, the comparison between the specific and regular PT leaves little room for distinction in the change of the measured parameters. A further RCT, examining the effect of the specialised versus the standard neck PT on the CD, revealed a similar situation [17]. Patients experienced an improvement in the TWSTRS score, but not between the groups. In the study by Tassorelli et al. PT was only performed ten times over two weeks [13]. It was possible to measure a decrease in pain, but not in dystonia-related symptoms or dystonia-specific quality of life. Perhaps, two weeks of therapy, even if intensive, is not sufficient. The study result of Hu et al. suggests that the multi-week therapy can also be applied as a home exercise programme [12]. A single guided therapy session was followed by six weeks of exercise five times a week. The focus on improving strength, range of motion and isometrics led to significant increases in the TWSTRS score. This is reflected financially in lower therapy costs [15]. The study by Boyce et al. underlines this therapeutic orientation [14]. They compared relaxation with active exercise therapy versus relaxation alone in the form of eight therapy sessions over twelve weeks. Due to the number of participants not achieved, but required by the power calculation, only a trend was achieved instead of a significant result. The use of kinesiotape, in addition to BoNT therapy, has, above all, a positive influence on the quality of life of CD patients [11]. The reduction of dystonia-related symptoms is predominantly achieved through BoNT therapy. A further pilot study determined that kinesiotaping reduces pain and even modulates sensory function, whereas sham taping had no effect [18]. The tape was applied for 14 days on the sternocleidomastoid muscle in two different directions. However, the study design was planned without BoNT therapy.

The combination with concomitant BoNT therapy allows for regarding most of the RCTs the better the comparability by measuring similar time points. The regular interval between the two injections is three months. The mandatory doctor’s visit thus enables a comparable study design.

The strength of this review is the strict inclusion of the RCTs examining the BoNT therapy and the concomitant PT to obtain valid results.

A limitation of the found RCTS, with the exclusion of Reference [15], is the small number of patients with 21 in the mean. Additional PT can optimize the different aspects of life and symptoms for patients suffering from CD, but it has to be examined in a large cohort for a clear recommendation. Therefore, a multicentre RCT with multimodal PT as an add-on treatment to BoNT is currently being implemented in Germany [19]. Moreover, only two studies compared the effect of BoNT and PT to BoNT alone [12,13]. Therefore, only here can the impact of physical activity be derived.

Furthermore, only two databases were checked for eligible studies in order to obtain the comparability with the previous systematic review [5]. Therefore, it is possible to overlook already existing RCTs on this topic, being listed in further databases.

## 5. Conclusions

This systematic review regarding additional PT to basic BoNT therapy shows a possible benefit for patients suffering from CD. While the effect of PT on posture is still up for debate; one can conclude that adjunctive PT might improve the quality of life of CD patients and may alleviate disability. However, there are no clear recommendations regarding concrete therapy intervention, its duration and frequency due to the paucity of studies reviewed and the small number of patients included in each study. Furthermore, a regular therapy session seems useful, independent of the continuous support by a physiotherapist or even after professional instruction with regard to a home exercise programme. Furthermore, CD is a chronic disorder; hence, a short-term therapy sequence does not seem appropriate

Therefore, research on the dose–response or dose–effect principle is required.

## Figures and Tables

**Figure 1 toxins-14-00784-f001:**
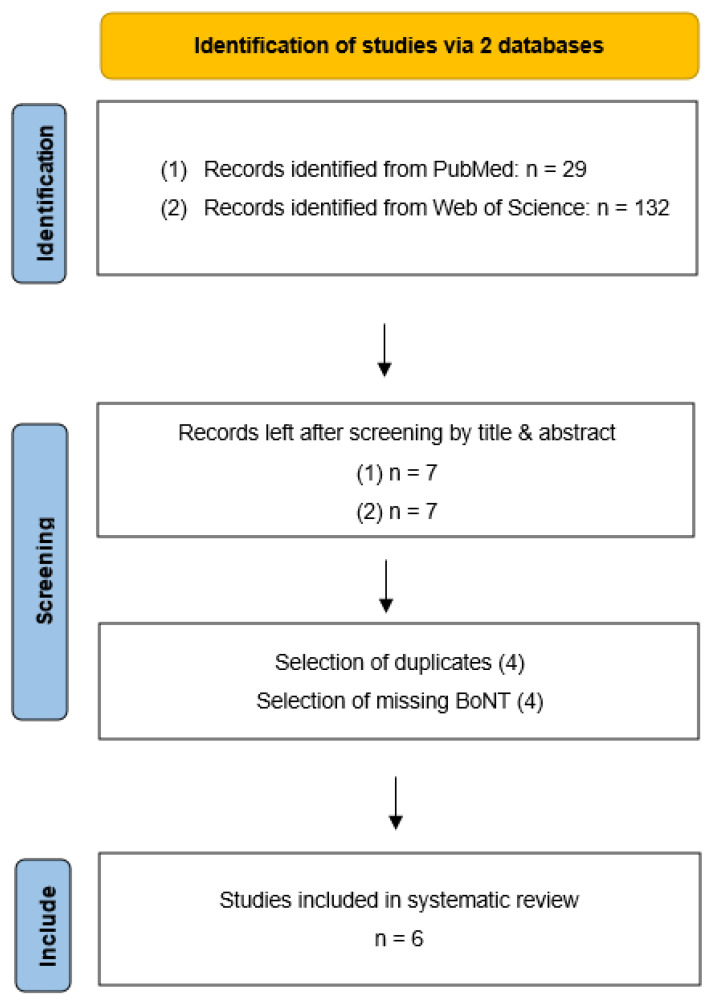
Flow chart of the literature search.

**Table 1 toxins-14-00784-t001:** Overview of the included RCTs.

Authors	Patients	Duration/Severity of CD	Medication BoNT-A	Intervention		Measuring Time Points	Results
				Control			
Dec-Cwiek et al. (2021)	*n* = 25 4 of which were females 54.7 ± 12.4	27.8 ± 12.4 years TWSTRS severity: 1 = 16.6 ± 9.01 2 = 19.11 ± 9.76 3 = 15.97 ± 8.79	yes	Crossover design: Change of experimental group after 12 weeks Taping: once a week for 4 weeks	1: BoNT + Kinesiotaping	Baseline 3 m 6 m 9 m	- Significant improvement in dystonia symptoms after BoNT injection regardless of group allocation - Quality of life improved significantly after taping
					2: BoNT + Sham taping		
					3: BoNT (control)		
Hu et al. (2019)	*n* = 16 9 of which were females 64.5 ± 5.4 (healthy controls: *n* = 10)	14.4 ± 10.9 years TWSTRS severity: BoNT = 22.2 ± 11.2 BoNT + PT = 18.4 ± 8.7	Yes (but patients who reported suboptimal benefits with BoNT alone)	BoNT alone (*n* = 8)		Baseline 6 w 3 m	- TWSTRS score (including pain), range of motion (ROM) and sensorimotor plasticity (SP) improved in PT-BoNT group - SP reached values of healthy controls
				BoNT + PT: 60 min of manual PT followed by six weeks of home-exercise programme/15 min/5× per week (*n* = 8)			
Van den Dool et al. (2019)	*n* = 96 59 of which were females 58.9 ± 9.2 (finished: *n* = 72, thereof 42 female)	12.7 ± 10.4 years TWSTRS severity: SPT: 16.91 ± 4.88 RPT = 16.00 ± 5.1	yes	SPT: specialised physical therapy (*n* = 40) Mean: 31 ± 11 PT sessions 6 weeks: 2× per week (30 min) After 6 weeks: 1× per week (6 months in total) After 6 months: 1× per month 5× per day/10–15 min home exercise		Baseline 6 m 12 m	- No significant between group differences after 12 months - Both groups improved significantly - SPT: higher patient perceived effects and general health perception, lower treatment costs
				RPT: regular physical therapy (*n* = 32) Mean: 41 ± 24 PT sessions 1× per week (12 month in total)			
Stankovic et al. (2017)	*n* = 14 11 of which were females 42.3 ± 5.6	13.5 ± 6.4 months	yes	BoNT + physical therapy (*n* = 9) PT five days after BoNT/five times weekly for two weeks		Baseline 1 m 3 m 6 m	- Highly significant increase in all the parameters of TWSTRS and decrease in changes in Tsui scale to all time points for BoNT + PT - Only PT with highly significant decrease of changes in Tsui scale was noticed only after one months
				Only physical therapy (*n* = 5)			
Boyce et al. (2012)	*n* = 20 14 of which were females 57.8 ± 7.8 years	10.2 ± 7.9 years TWSTRS: IG = 37.7 ± 10.7 CG = 34.7 ± 13.4	None = 13 BoNT = 7	Experimental group (*n* = 9) - Relaxation and individualised exercise program of active exercises to increase muscle strength of antagonist muscles and induce normal head posture - Combined with BoNT-A (*n* = 3) - 12 weeks program with 8 supervised sessions (30 min) (first 4 weeks: 1× per week and daily home exercises and the following 8 weeks: 1× per two weeks)		Baseline 6 w 3 m 4 m	- Patients were able to perform an active exercise program, but there was only a trend towards better scores on the TWSTRS and decreased depression (Beck Depression Inventory) - A power calculation showed the need to include 34 patients and an improvement of seven points on the TWSTRS for a positive treatment effect
				Control group (*n* = 11) - Whole body relaxation program - Combined with BoNT-A (*n* = 4)			
Tassorelli et al. (2006)	*n* = 40 13 of which were females 51.3 ± 15.6 years	11.5 ± 2.9 years	yes	- *n* = 20 - BoNT-A combined with physiotherapy: massage, stretching, postural re-education, strengthening of the axial muscles and 30 min biofeedback - 10 sessions of 60–90 min for two weeks		Baseline 3 m 6 m 9 m	- No significant decrease of severity of dystonia in both groups in the Tsui scale and TWSTRS - Significant less pain and increased ADL functioning in PT-group - The effects of BoNT-A lasted significantly longer, and a significantly lower dose was needed for the next injection
				- *n* = 20 - BoNT alone			

BoNT = botulinum neurotoxin; CD = cervical dystonia; PT = physiotherapy; *n* = number; TWSTRS = Toronto Western Spasmodic Torticollis Rating Scale; SP = sensorimotor plasticity; SPT = specialised PT; RPT = regular PT.

**Table 2 toxins-14-00784-t002:** Evaluation of study quality by PEDro scale.

PEDro Scale	Dec-Cwiek et al. (2021) [11]	Hu et al. (2019) [12]	van den Dool et al. (2019) [15]	Stankovic et al. (2017) [16]	Boyce et al. (2012) [14]	Tassorelli et al. (2006) [13]
1. Eligibility criteria	y	y	Y	n.a.	y	y
2. Randomly allocated	y	y	Y	y	y	y
3. Allocation concealed	n	n	Y	n	Y	n
4. Similar at baseline	y	y	y	y	Y	y
5. Blinding of subjects	y	n	n	n	n	n
6. Blinding of therapists	n	n	n	n	n	n
7. Blinding of assessors	y	y	y	y	y	n
8. One key outcome of 85%	n	y	n	y	y	y
9. Allocated treatment/ITT	y	y	y	y	y	y
10. Between-group statistical comparison	y	y	y	y	y	y
11. Point measures and measures of variability	y	y	y	n	y	y
	**good**	**good**	**good**	**good**	**good**	**good**

Y = yes; y= yes; n = no; n.a. = not announced/applicable. Cut-points: 9–10 = excellent; 6–8 = good; 4–5 = fair; <4 = poor. The colour is for better recognition of evaluation

**Table 3 toxins-14-00784-t003:** Evaluation of the risk of bias.

Study ID	D1	D2	D3	D4	D5	Overall		
Dec-Cwiek et al.(2021) [11]	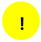				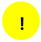	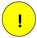		Low risk
Hu et al. (2019) [12]	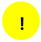					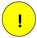	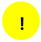	Some concerns
van den Dool et al. (2019) [15]						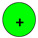	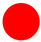	High risk
Stankovic et al. (2017) [16]	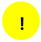				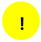	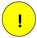		
Boyce et al. (2012) [14]		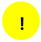				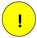	D1	Randomisation process
Tassorelli et al. (2006) [13]	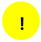				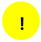	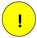	D2	Deviations from the intended interventions
							D3	Missing outcome data
							D4	Measurement of the outcome
							D5	Selection of the reported result

Green circle with + = low risk, Yellow circle with ! = some concerns, Red circle = a template and shows that the references have no high risk.

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
