# Peer review of "Physiotherapy for Cervical Dystonia: A Systematic Review of Randomised Controlled Trials"

_toxins, 2022, doi:10.3390/toxins14110784_

Round 1
Reviewer 1 Report
The paper "Physiotherapy for cervical dystonia: a systematic review of randomized controlled trials" is a review on the effect of physical therapy (PT) either as a stand-alone therapy of cervical dystonia, or as an add-on therapy for patients treated with botulinum toxin.
I have few concerns regarding this paper:
The paper chose to review only randomized controlled studies and thereby reduced the number of papers down to 6 which is scant. From those 6 studies, 5 had small samples and the conclusions from this review should be interpreted with great caution.
The study of Hu (n=16) showed improvements in TWSTRS subsets of pain, disability severity and range of motion (ROM).
The study of Tassorelli (n=40) showed that adjunctive PT improved pain and ADL functioning and showed a longer BT effect.
One study (DecCwiek et al., n=25) compared BT+ kinesiotaping to BT alone and found beneficial effect of kinesiotaping on quality of life.
One study (Van den Dool et al., n=96) compared BT+ regular PT to specialized PT, without BT alone as a reference. Both sorts of PT improved over time in TWSTRS disability score, quality of life, depression and anxiety compared to base-line.
One study (Stankovic et al., n=14) compare BT+PT to PT alone.
One study (Boyce et al., n=20) was very problematic, including both CD patients treated and untreated with BT, which compared exercise + relaxation therapy vs. relaxation therapy only
It is of note that only two studies (!!!) (Hu et al., Tassorelli et al.) compared BT+PT to BT alone.
The 'limitation' section is too concise. The authors should note that only 2 studies compared BT+PT vs. BT alone. From the other studies, it is not clear what the impact of physical activity is for CD. They must add a note that no firm statement or conclusion can be made due to the paucity of studies reviewed and the small number of patients included in each study.
Table 1 is confusing. I would suggest that the column "Intervention-control" be divided into 2 columns: "Design" and "Intervention".
Table 2 and Table 3 reflect more of the same. I think that Table 3 is redundant.
Conclusion
I think the conclusion should be more conservative. I would state something like: "from this review one may suggest that PT as an adjunctive therapy to BT for CD, is possibly beneficial. While the effect of PT on posture is still on debate, we may suggest/state/conclude that adjunctive PT might improve the quality of life of CD patients, and perhaps may alleviate disability".
Minor comment:
The paper is written carelessly with many typo errors. For example:
Table 1: Typo in the row of Van den Dool et al. ("betwwn"). In the row of Stancovic "fort wo weeks".
The row of Boyce et al. is unclear for the most part.
The row of Hu: TWSTRS improved in the pain score as well as ROM: it is worth to note!
Table 2 is wrongly labeled as Table 1.
Reviewer 2 Report
This is a potentially interesting paper, reporting on a systematic review of randomized controlled trials regarding Physiotherapy for cervical dystonia. The strength of this study is the adequate methodology, but as expected the different study design, the variability in the outcome measures, and most importantly the low number of patients enrolled in each study did not allow the authors to draw any final conclusion on the effectiveness and cost-efficacy of physiotherapy in this common type of dystonia. No novel concept is therefore added to this important topic, after the De Pauw et al review published in 2014.
In addition, the manuscript requires a full language revision also taking into consideration some wrong name usage in English (e.g. Line 190: Musculus sternocleidomastoideus ? OR in Table 1, patients: better write n = 25,
11 of which were females etc.).
